# Anthropometric Measurements and Admission Parameters as Predictors of Acute Respiratory Distress Syndrome in Hospitalized COVID-19 Patients

**DOI:** 10.3390/biomedicines11041199

**Published:** 2023-04-18

**Authors:** Vladimir Zdravković, Đorđe Stevanović, Neda Ćićarić, Nemanja Zdravković, Ivan Čekerevac, Mina Poskurica, Ivan Simić, Vladislava Stojić, Tomislav Nikolić, Marina Marković, Marija Popović, Ana Divjak, Dušan Todorović, Marina Petrović

**Affiliations:** 1Department of Interventional Cardiology, Cardiology Clinic, University Clinical Center Kragujevac, 34000 Kragujevac, Serbia; 2Department of Internal Medicine, Faculty of Medical Sciences, University of Kragujevac, 34000 Kragujevac, Serbia; 3Department of Pathophysiology, Faculty of Medical Sciences, University of Kragujevac, 34000 Kragujevac, Serbia; 4Pulmonology Clinic, University Clinical Center Kragujevac, 34000 Kragujevac, Serbia; 5Department of Medical Statistics and Informatics, Faculty of Medical Sciences, University of Kragujevac, 34000 Kragujevac, Serbia; 6Urology and Nephrology Clinic, University Clinical Center Kragujevac, 34000 Kragujevac, Serbia; 7Center of Medical Oncology, University Clinical Center Kragujevac, 34000 Kragujevac, Serbia; 8Department of Physical Medicine and Rehabilitation, University Clinical Center Kragujevac, 34000 Kragujevac, Serbia; 9Department of Physical Medicine and Rehabilitation, Faculty of Medical Sciences, University of Kragujevac, 34000 Kragujevac, Serbia; 10Department of Ophtamology, Faculty of Medical Sciences, University of Kragujevac, 34000 Kragujevac, Serbia; 11Ophtalmology Clinic, University Clinical Center Kragujevac, 34000 Kragujevac, Serbia

**Keywords:** ARDS, bioelectrical impedance analysis, body fat percentage, COVID-19, obesity

## Abstract

**Aim**: We aimed to single out admission predictors of acute respiratory distress syndrome (ARDS) in hospitalized COVID-19 patients and investigate the role of bioelectrical impedance (BIA) measurements in ARDS development. **Method**: An observational, prospective cohort study was conducted on 407 consecutive COVID-19 patients hospitalized at the University Clinical Center Kragujevac between September 2021 and March 2022. Patients were followed during the hospitalization, and ARDS was observed as a primary endpoint. Body composition was assessed using the BMI, body fat percentage (BF%), and visceral fat (VF) via BIA. Within 24 h of admission, patients were sampled for blood gas and laboratory analysis. **Results**: Patients with BMI above 30 kg/m^2^, very high BF%, and/or very high VF levels were at a significantly higher risk of developing ARDS compared to nonobese patients (OR: 4.568, 8.892, and 2.448, respectively). In addition, after performing multiple regression analysis, six admission predictors of ARDS were singled out: (1) very high BF (aOR 8.059), (2) SaO_2_ < 87.5 (aOR 5.120), (3) IL-6 > 59.75 (aOR 4.089), (4) low lymphocyte count (aOR 2.880), (5) female sex (aOR 2.290), and (6) age < 68.5 (aOR 1.976). **Conclusion**: Obesity is an important risk factor for the clinical deterioration of hospitalized COVID-19 patients. BF%, assessed through BIA measuring, was the strongest independent predictor of ARDS in hospitalized COVID-19 patients.

## 1. Introduction

Severe acute respiratory syndrome coronavirus 2 (SARS-CoV-2) is responsible for a significant number of deaths and health impairments worldwide [1]. The illness is characterized mostly by respiratory clinical manifestations ranging from mild to severe clinical forms [2]. However, in 10–20% of patients, the disease is complicated by respiratory failure, acute respiratory distress syndrome (ARDS), and multisystem organ failure [3,4].

Despite continuous research efforts, the exact mechanisms of clinical deterioration are not fully elucidated. In a practical manner, it would be beneficial to single out conditions and clinical characteristics associated with ARDS development in order to construct prediction models and promptly single out patients at risk of clinical deterioration. According to the available literature, many parameters are associated with COVID-19 ARDS, such as older age, male sex, certain comorbidities, smoking, impaired gas exchange, elevated biomarkers of inflammation, and impaired coagulation [5,6,7,8,9,10,11]. However, the exact selection of predictors and their impact on ARDS development in hospitalized COVID-19 patients are not uniform across the literature, mostly due to the significant heterogeneity of cohort characteristics and methodological approach.

Obesity, according to body mass index (BMI), is associated with disease severity and the need for ICU treatment (OR 1.2–5.5) [12,13,14,15]. However, BMI is a basic anthropometric measurement that neglects body composition and the presence of adipose tissue. Given that most of the pathophysiological mechanisms through which obesity possibly affects the COVID-19 course are the effect of adipose tissue, assessing body composition solely using the BMI could be suboptimal [16,17,18].

Therefore, we aimed to single out admission predictors of ARDS in hospitalized COVID-19 patients, based on sociodemographic and medical history data, as well as blood gas and laboratory analysis on admission. Additionally, we aimed to examine the impact of obesity on COVID-19 ARDS, as well as compare BMI with anthropometric measurements given by BIA method, such as total body and visceral fat. To the best of the authors’ knowledge, this is the first registry regarding BIA measurements and COVID-19 outcomes.

## 2. Materials and Methods

### 2.1. Study Population

An observational, prospective cohort study included 407 consecutive COVID-19 patients hospitalized at the University Clinical Center Kragujevac between September 2021 and March 2022. The study was granted approval by the University Clinical Center Kragujevac (Serbia) Ethical committee.

Our COVID-19 center consisted of standard, semi-intensive, and intensive care units focused on severe to critically ill patients [19]. Patients were followed during the hospitalization, and ARDS development was observed as a primary endpoint. The ARDS is an acute onset clinical syndrome characterized by diffuse, bilateral alveolar damage, inflammation, and edema, resulting in respiratory failure [20]. ARDS was defined using the Berlin criteria [21]: (1) timing within 1 week of clinical insult or new/worsening respiratory symptoms; (2) bilateral opacities seen on chest imaging, not fully explained by effusions, lobar/lung collapse, or nodules; (3) respiratory failure not fully explained by cardiac failure/fluid overload; (4) oxygenation impairment, defined as the ratio of partial pressure of arterial oxygen (PaO_2_) to the fraction of inspired oxygen (FiO_2_) below 300 mm Hg.

Inclusion and exclusion criteria are presented in Appendix A. We note that, during the data collection period, 1240 patients were hospitalized in our COVID-19 center; however, 825 patients were excluded according to exclusion criteria. Since there was an insufficient number of underweight patients (total of eight patients according to either BMI or BF% measurement), those patients were additionally excluded from further analysis.

### 2.2. Data Collection

Sociodemographic and medical history data were obtained using patients’ electronic medical records (Health Informational System version 2, ComTrade, Kragujevac, Serbia). Arterial and peripheral venous blood for further analysis was routinely sampled at the time of admission.

Within 48 h of hospital admission, patients were measured on the TANITA BC-543 device (Tanita Corporation, Tokyo, Japan). According to the manufacturer’s instructions, patients were measured in the morning, barefoot, in light clothing, before the first meal. In order to minimize the effect of acute infection and fever, patients were measured in an afebrile state.

Anthropometric parameters of interest were the following:(I)BMI, calculated using the formula BMI [kg/m^2^] = BM [kg]/BH^2^ [m^2^], where BM is the body mass expressed in kg (with 0.1 kg precision), and BH is body height expressed in m (with 0.01 m precision). According to BMI values, patients were categorized as (I) underweight < 18.5 kg/m^2^, (II) normal weight 18.6–24.9 kg/m^2^, (III) overweight 25–29.9 kg/m^2^, (IV) class 1 obesity 30–34.9 kg/m^2^, (V) class 2 obesity 35–39.9 kg/m^2^, or (VI) class 3 obesity > 40 kg/m^2^ [22].(II)Body fat percentage (BF%), expressed as a percentage of the total mass (with 0.1% precision). According to BF% values, regarding age and sex, patients were categorized as (I) low BF%, (II) normal BF%, (III) high BF%, or (IV) very high BF% (age- and sex-adjusted cutoff values are presented in Table 1) [23].(III)Visceral fat (VF) level, according to which patients were categorized as (I) normal (1–9), (II) high (10–14), or (III) very high (≥15) [24].

### 2.3. Statistical Analysis

Statistical analysis was performed using the SPSS statistical package (version 25.0, IBM corporation, Armonk, NY, USA). Differences in quantitative data were tested using the Mann–Whitney U test. If applicable, continuous data were further transformed into a binary variable. When dividing continuous variables into categories, an accepted reference line or cutoff values given by ROC analysis (Appendix A) were used, depending on clinical applicability. After identifying the variables associated with primary outcome, uni- and multivariable binary logistic regression was performed. The strength of the relationship between examined variables and outcome was expressed as the odds ratio (OR) with 95% confidence interval (95% CI) for univariate, and as the adjusted odds ratio (aOR) with 95% CI for multivariate analysis. A *p*-value < 0.05 was considered significant.

## 3. Results

Our cohort consisted of 407 adult COVID-19 patients hospitalized at the University Clinical Center Kragujevac (Serbia), whose characteristics are presented in Table 2. During the hospitalization, 98.5% of patients required some form of oxygen support, 51.6% of patients required noninvasive/invasive ventilation, and 35.1% of patients developed ARDS, with a mortality rate of 17.4%. Younger age and female sex were associated with ARDS development, while patients with and without ARDS did not significantly differ in terms of comorbidities. In addition, patients who developed ARDS had a shorter period between disease onset and the need for inpatient treatment, more frequently required oxygen support upon admission, had a longer hospital stay, and had a higher mortality rate.

The prevalence of obesity was 37.9% according to BMI and 49.6% according to BF%, while 38.6% of patients had an excessive level of VF. (Figure 1) Furthermore, patients who developed ARDS more frequently had BMI above 30 kg/m^2^ (*p* < 0.001), as well as very high body fat percentage (*p* < 0.001) and visceral fat level (*p* < 0.001).

The admission blood gas and laboratory analysis is presented in Table 3. Patients with ARDS, compared to those without, had significant differences in blood cell count (lower lymphocyte and higher granulocyte count), more impaired gas exchange (lower values of oxygen saturation (SaO_2_) and partial oxygen pressure (PaO_2_)), and more increased inflammatory biomarkers (including C-reactive protein (CRP), lactate dehydrogenase (LDH), and interleukin-6 (IL-6)).

After performing the univariate analysis, 11 variables were selected for the final binary logistic regression model. (Table 4) Six variables gave statistically significant contributions to the model: (1) very high BF (aOR 8.059), (2) SaO_2_ < 87.5 (aOR 5.120), (3) IL-6 > 59.75 (aOR 4.089), (4) low lymphocyte count (aOR 2.880), (5) female sex (aOR 2.290), and (6) age < 68.5 years (aOR 1.976). The score was statistically significant (c^11^ = 178.54; *p* < 0.001), with a sensitivity of 76%, specificity of 87.7%, and C-index of 0.885 (*p* < 0.001) (Figure 2).

## 4. Discussion

This study aimed to examine the impact of obesity on ARDS development and the likelihood of variables available upon admission to predict ARDS in hospitalized COVID-19 patients. Patients who developed ARDS had a shorter time from disease onset to hospital admission, and, with each day that passed from disease onset to hospital admission, the risk of ARDS development decreased by 7.7% (OR 0.923) [25].

The majority of patients had disturbed nutritional status according to all three anthropometric measurements (BMI, BF%, and VF). Body composition disturbances in hospitalized COVID-19 patients are not unexpected. Firstly, the high prevalence of obesity and obesity-related comorbidities has reached pandemic proportions [22], with consequences also noticeable in Serbia. According to research from 2006, 55.7% of the adult population in Serbia was overweight or obese [26]. Secondly, several studies have demonstrated that obesity is a significant risk factor for hospital admission [12,13,14,15]; therefore, it is somewhat expected to have a high prevalence of obesity among hospitalized COVID-19 patients. Furthermore, patients with BMI above 30 kg/m^2^, very high BF%, and/or very high VF levels were at significantly higher risk of developing ARDS compared to nonobese patients (OR: 4.568, 8.892, and 2.448, respectively), which is in accordance with most of the literature data [12,13,14,18,27,28,29,30,31,32]. Obesity augments the risk of critical illness and mortality in SARS-CoV-2 infection through multiple mechanisms, including chronic inflammation, immune dysregulation, respiratory compromise, impaired pulmonary function, and endocrine and endothelial dysfunction [16,17,33]. However, the majority of mechanisms through which obesity can initiate the clinical deterioration of COVID-19 patients, especially in terms of inflammation and immune response dysregulation, are the effect of the adipose tissue. In obese patients, adipose tissue is responsible for chronic low-grade inflammation, characterized by increased production of cytokines and acute-phase reactants, as well as the macrophage accumulation, resulting in both innate and adaptive immune response impairment. Additionally, chronic inflammation in obese patients could potentially enhance the inflammatory response to COVID-19, resulting in a hyperinflammatory state and disease progression [16,17,34]. This could potentially explain why BF% was a stronger predictor of ARDS development compared to BMI, and why a very high BF% was the strongest independent predictor of ARDS (aOR 8.059) in a multiple regression analysis.

Interestingly, the VF level did not maintain statistical significance in a final regression model. This result could partially be explained by the inadequacy of BIA visceral fat assessment to differentiate adipose tissue between visceral and abdominal compartments [35].

In addition to obesity, the most common comorbidities in our cohort were arterial hypertension, diabetes mellitus, and chronic kidney disease. Despite literature data linking several comorbidities with clinical deterioration and worse outcomes in COVID-19 patients, no comorbidity showed a significant impact on ARDS development in our research [36]. Such differences could partially be explained by the characteristics of the comorbidities in the examined cohort, such as their prevalence, severity, duration, medication burden, and presence of disease-related complications. Therefore, the presence of well-controlled, uncomplicated comorbidities in our cohort could be a possible explanation for the absent impact of the most frequent comorbidities on the ARDS development. Additionally, patients initially hospitalized or requiring hospital treatment for non-COVID-19 pathology, with terminal stage of malignant disease, and with end-stage of renal disease were excluded from our study.

Among demographic characteristics, the female sex was shown to be an independent predictor of ARDS (aOR 2.290). Although the male sex is commonly linked with a higher risk of ICU admission and mortality in hospitalized COVID-19 patients [37,38], some explanations could include different cohort characteristics, more present high-risk behavior related to COVID-19, higher frequency of pre-existing comorbidities, differences in innate immune response, and different activity and expression of ACE 2 [39,40]. In addition, literature data advocate older age as ARDS and mortality risk factors in hospitalized COVID-19 patients [5,10,41]. However, our study singled out age below 68 years to be an independent predictor of ARDS (aOR 1.976). A possible explanation for such a result could be that older patients had a significantly higher mortality rate compared to those younger than 68 years (24.5% and 11.4%, respectively; *p* = 0.001). Thus, those patients experienced lethal outcome in greater numbers before ICU admittance and ARDS development/diagnosis.

With regard to gas exchange upon admission, patients who developed ARDS during the course of inpatient treatment more frequently required oxygen support upon admission and had more prominent hypoxemia, assessed as lower values of PaO_2_ and SaO_2_. Additionally, low SaO_2_ (≤87%) was the second strongest predictor of ARDS in the multiple regression analysis model. Such findings are of significant practical importance, since both PaO_2_ and SaO_2_ are affordable, widely used, and feasible indicators of hypoxemia’s severity. Pronounced hypoxemia upon admission has been confirmed to be a significant risk factor for COVID-19 progression and lethal outcome [5,7,42]. The combined effects of capillary damage associated with COVID-19 affect tissue oxygenation in key organs. A vicious cycle ensues, as inflammation and hypoxia cause capillary function to deteriorate, which accelerates hypoxia-induced inflammation, oxidative stress, tissue damage, and ARDS [43].

With regard to laboratory analysis upon admission, patients with and without ARDS differed in several parameters. Firstly, patients who developed ARDS had significantly higher values of inflammatory biomarkers upon admission, including IL-6, CRP, and LDH. Furthermore, IL-6 > 59.75 pg/mL upon admission was an independent predictor of ARDS with an aOR of 4.089. A pronounced inflammatory response is considered to be one of the keystones of COVID-19 progression, and elevated inflammatory biomarkers are one of the most commonly advocated predictors of ICU admission and death [10,44,45]. Secondly, patients significantly differed in blood cell count. Clinical observation of lymphopenia has been apparent since the start of the COVID-19 pandemic and may be associated with worsening disease [46]. Our results are in accordance with the literature in that a low lymphocyte count in our cohort was singled out as an independent predictor of ARDS development (aOR 2.880).

Lastly, we singled out independent predictors of ARDS development and provided an ARDS risk assessment score relying on the parameters available upon hospital admission: y = 0.007 + (very high BF%) × 8.059 + (SaO_2_ < 87.5%) × 5.120 + (IL6 > 59.75 pg/mL) × 4.089 + (low lymphocyte count) × 2.880 + (female sex) × 2.290 + (age < 68.5 years) × 1.976. The model had a C-index of 0.885 (*p* < 0.001), indicating a strong predictive model. Due to its high specificity (87.7%) and negative predictive value (86.6%), this model could have clinical importance as a “rule-out” tool in helping physicians define patients with a high risk of developing ARDS. Selecting predictors of worse outcomes and understanding the mechanisms responsible for the clinical deterioration of hospitalized COVID-19 patients has been the focus of scientific research for the past 2 years. More than 30 different parameters can be linked with ARDS development in COVID-19, and several risk assessment scores can be found [5,6,7,8,9,10,11,47,48,49]. However, the selection of predictors, as well as their cutoff and aOR values, significantly varies across the literature. Additionally, proposed predictive models differ in terms of accuracy (C-index according to ROC analysis), sensitivity, and specificity, which can greatly influence their potential utilization. These variations can be explained by differences in the statistical and methodological approach, patient characteristics, availability of examined parameters, health system organization, SARS-CoV-2 variant predominance, etc. Therefore, no specific model should be generally accepted irrespective of geographical, sociodemographic, patient structure, hospital equipment, and other important specifics. Instead, such research tends to extend the data to understand the mechanisms of COVID-19 progression and to point out that, in different settings, physicians can select patients with a higher risk of disease progression and death. In addition, to the authors’ knowledge, this is the first registry to use admission BIA measurements in an ARDS prediction model. Despite their limitations, available and affordable BIA measurements gave the strongest independent predictor of ARDS development.

Our study had certain limitations. Firstly, active infection can potentially affect the BIA measurement, especially in the febrile state [50]. This limitation could, therefore, have been responsible for misinterpreting the body composition, in terms of a higher percentage of body fat. In order to minimize this effect, patients were measured within the first 48 h upon admission, in the absence of fever. Secondly, although BIA measurements provide satisfactory insight into total body fat and fat-free mass, this method has difficulties in distinguishing visceral from abdominal fat, for which computed tomography and magnetic resonance imaging remain the gold standard [35,51]. This could have been responsible for underestimating the individual impact of VF levels in our cohort. Lastly, the number of patients included may be insufficient for the generalization of the results. Therefore, we encourage fellow physicians and researchers to continue investigating BIA measurements in predicting COVID-19 outcome.

## 5. Conclusions

Obesity is an important risk factor for the clinical deterioration of hospitalized COVID-19 patients. Because of the insight into the total body and visceral fat, BIA measurements could be a useful and affordable tool in selecting COVID-19 patients with a high risk of developing ARDS.

## Figures and Tables

**Figure 1 biomedicines-11-01199-f001:**
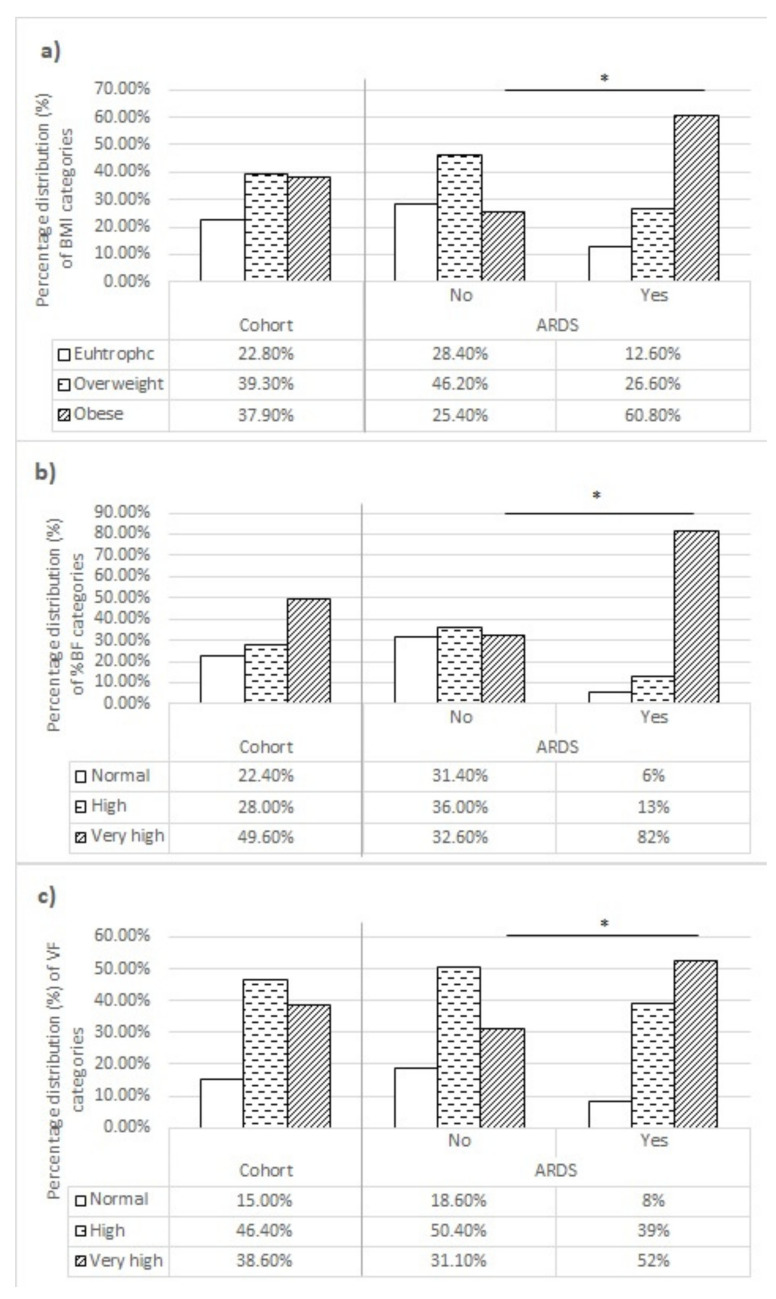
Percentage distribution of body composition categories for (**a**) BMI, (**b**) %BF, and (**c**) VF level, with regard to ARDS development. Abbreviations: ARDS—acute respiratory distress syndrome; BMI—body mass index; VF—visceral fat; %BF—body fat percentage. * Statistical significance at <0.05.

**Figure 2 biomedicines-11-01199-f002:**
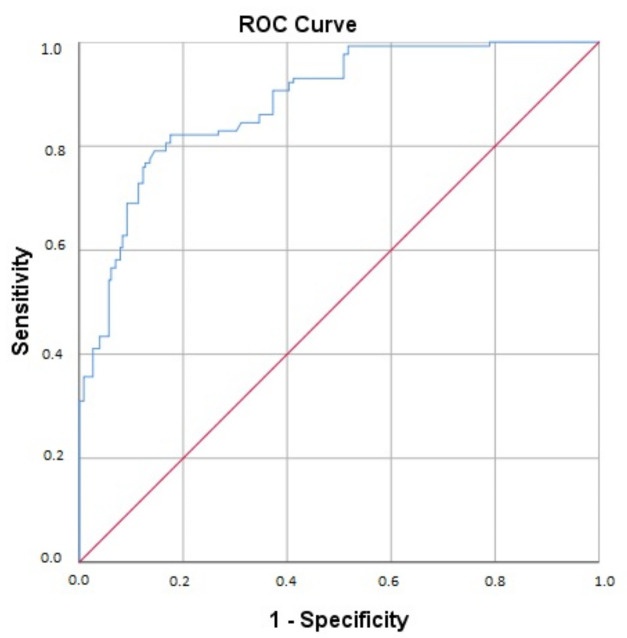
Receiver operator characteristics (ROC) curve of multiple regression analysis model in predicting ARDS. Legend: Blue line—ROC curve, red line—diagonal reference line.

**Table 1 biomedicines-11-01199-t001:** Age- and sex-adjusted cutoff values for BF% categories.

Sex	Age (Years)	BF% Categories			
		Low	Normal	High (Overweight)	Very High (Obesity)
Female	20–39	<21%	21–32.9%	33–39.5%	>39.5%
	40–59	<23%	23–33.9%	34–40%	>40%
	≥60	24%	24–35.9%	36–41.5%	>41.5%
Male	20–39	<7%	7–19.9%	20–25%	>25%
	40–59	<10.5%	10.5–21.9%	22–27.5%	>27.5%
	≥60	<12%	12–24.9%	25–30%	>30%

Abbreviations: BF%—Body fat percentage. BF% and VF levels were assessed using BIA analysis, by measuring the tissue’s impedance to a low current electrical impulse.

**Table 2 biomedicines-11-01199-t002:** Demographic and medical history data, with regard to ARDS development.

Cohort Characteristics		Frequency (Number of Cases) or Median Value (with IQR)			*p*-Value
		Cohort	No ARDS	ARDS	
**Age [years]**		68.0 (IQR 17.0)	68.5 (IQR 16.0)	66.0 (IQR 19.0)	0.006 *
**Sex**	Male	62.9% (*n* = 256)	68.9% (*n* = 182)	51.7% (*n* = 74)	0.001 *
	Female	37.1% (*n* = 151)	31.1% (*n* = 82)	48.3% (*n* = 69)	
**COMORBIDITIES**					
**Arterial hypertension**		68.1% (*n* = 277)	70.5% (*n* = 186)	63.6% (*n* = 91)	0.182
**Diabetes mellitus**		26.5% (*n* = 108)	25.4% (*n* = 67)	28.7% (*n* = 41)	0.482
**Chronic kidney disease**		14.7% (*n* = 60)	14.8% (*n* = 39)	14.7% (*n* = 21)	1.000
**Neurological condition ^1^**		3.4% (*n* = 14)	3.8% (*n* = 10)	2.8% (*n* = 4)	0.778
**Previous myocardial infarction**		3.9% (*n* = 16)	3.8% (*n* = 10)	4.2% (*n* = 6)	0.797
**Malignancy**		6.1% (*n* = 25)	7.2% (*n* = 19)	4.2% (*n* = 6)	0.283
**Obstructive lung disease ^2^**		3.2% (*n* = 13)	3.4% (*n* = 9)	2.8% (*n* = 4)	1.000
**Charlson Comorbidity Index**		3.0 (IQR 2.0)	3.0 (IQR 2.0)	3.0 (IQR 3.0)	0.052
**DISEASE COURSE AND OUTCOME**					
**Time from disease onset to hospital admission [days]**		6.0 (IQR 6.0)	7.0 (IQR 7.0)	6.0 (IQR 5.0)	0.003 *
**ARDS development**		35.1% (*n* = 143)	/	/	/
**Mortality**		17.4% (*n* = 71)	9% (*n* = 24)	44.1% (*n* = 63)	<0.001 *
**Hospital stay [days]**		17.0 (IQR 11.0)	14.5 (IQR 9.75)	21.0 (IQR 14.0)	<0.001 *
**Oxygen support requirement on admission**		92.9% (*n* = 378)	90.5% (*n* = 239)	97.2% (*n* = 139)	0.014 *

Abbreviations: ARDS—acute respiratory distress syndrome; IQR—interquartile range. * Statistical significance level at <0.05. ^1^ Neurological condition: the presence of history of stroke, brain tumor or malformation, vascular disease, dementia of any etiology, etc. ^2^ Obstructive lung disease: the presence of either chronic obstructive lung disease or bronchial asthma.

**Table 3 biomedicines-11-01199-t003:** Admission blood gas and laboratory analysis, with regard to ARDS development.

Blood Gas and Laboratory Analysis	Median Values (IQR)			*p*-Value
	Cohort	No ARDS	ARDS	
PaO_2_ [kPa]	6.9 (IQR 1.4)	14.5 (IQR 9.75)	6.4 (IQR 1.2)	<0.001 *
SaO_2_ [%]	88 (IQR 6)	89 (IQR 5)	86 (IQR 5)	<0.001 *
WBC [10^9^/L]	8.2 (IQR 4.8)	8.2 (IQR 4.5)	8.3 (IQR 6.6)	0.751
Granulocyte count [10^9^/L]	6.9 (IQR 4.9)	6.5 (IQR 4.85)	7.4 (IQR 4.1)	0.710
Granulocytes [%]	84.2 (IQR 11.7)	83 (IQR 12.2)	85.9 (IQR 10.8)	0.001 *
Lymphocyte count [10^9^/L]	0.7 (IQR 0.5)	0.73 (IQR 0.57)	0.69 (IQR 0.43)	0.009 *
Lymphocytes [%]	9.1 (IQR 8.5)	9.5 (IQR 8.6)	8.5 (IQR 6.4)	0.036 *
RBC [10^12^/L]	4.5 (IQR 0.7)	4.48 (IQR 0.81)	4.47 (IQR 0.69)	0.968
HGB [g/L]	134 (IQR 21)	134.5 (IQR 21.75)	133 (IQR 21)	0.081
PLT [10^9^/L]	201 (IQR 113)	203.5 (IQR 108)	199 (IQR 115)	0.771
INR	1.08 (IQR 0.18)	1.08 (IQR 0.16)	1.07 (IQR 0.19)	0.079
aPTT [s]	31.4 (IQR 6.85)	31.5 (IQR 7)	33.3 (IQR 6.3)	0.917
Fibrinogen [g/L]	6.5 (IQR 2.15)	6.26 (IQR 2.06)	6.22 (IQR 2.09)	0.485
D-dimer [ug/mL]	0.93 (IQR 0.97)	0.97 (IQR 1.25)	0.85 (IQR 0.76)	0.159
Albumin [g/L]	36 (IQR 4)	36 (IQR 4)	36 (IQR 5)	0.337
AST [IU/L]	42 (IQR 35)	41 (IQR 31)	43 (IQR 41)	0.271
ALT [IU/L]	36 (IQR 36)	36 (IQR 38.25)	41 (IQR 40)	0.830
GGT [IU/L]	41 (IQR 65)	42 (IQR 65)	41 (IQR 64)	0.819
BUN [mmol/L]	7.8 (IQR 5.3)	8.4 (IQR 5.5)	7.1 (IQR 4.2)	0.054
Creatinine [mmol/L]	92 (IQR 41)	91 (IQR 42)	94 (IQR 40)	0.216
LDH [U/L]	773 (IQR 382)	702 (IQR 388)	890 (IQR 242)	<0.001 *
Ferritin [ug/L]	838 (IQR 745)	815 (IQR 792)	877 (IQR 746)	0.335
CK [U/L]	107(IQR 166)	91.5 (IQR 152.75)	156 (IQR 179)	0.001 *
CKMB [U/L]	18 (IQR 10)	18 (IQR 9)	18 (IQR 11)	0.386
CRP [mg/L]	99 (IQR 96.1)	96.6 (IQR 92.5)	108.7 (IQR 99.8)	0.009 *
PCT [ng/mL]	0.11 (IQR 0.18)	0.1 (IQR 0.17)	0.114 (IQR 0.18)	0.413
cTnI [ng/mL]	0.0038 (IQR 0.014)	0.00145 (IQR 0.0145)	0.006 (IQR 0.0127)	0.637
pro-BNP [pg/mL]	559 (IQR 880)	604 (IQR 959)	389 (IQR 895)	0.066
IL-6 [pg/mL]	58.7 (IQR 97)	47.4 (IQR 89.15)	89.8 (IQR 119.3)	<0.001 *

Abbreviations: ALT—alanine transaminase; aPTT—activated partial thromboplastin clotting time; AST—aspartate transaminase; BUN—blood urea nitrogen; CK—creatine kinase; CKMB—muscle–brain form of creatine kinase; CRP—C-reactive protein; D—D-dimer; GGT—gamma-glutamyl transferase; Gran—granulocyte; Hgb—hemoglobin; hsTnI—high-sensitivity troponin I; IL-6—interleukin 6; INR—international normalized ratio; LDH—lactate dehydrogenase; Lym—lymphocytes; NT pro-BNP—N-terminal pro-brain natriuretic peptide; PaO_2_—partial pressure of oxygen; PCT—procalcitonin; PLT—platelets; RBC—red blood cells; SaO_2_—oxygen saturation of blood; WBC—white blood cells. * Statistical significance level at <0.05.

**Table 4 biomedicines-11-01199-t004:** Crude and adjusted OR for variables available upon hospital admission with regard to predicting ARDS development of hospitalized COVID-19 patients.

Variable		Frequency of ARDS	Crude OR		Adjusted OR	
			OR (95% CI)	*p*-Value	OR (95% CI)	*p*-Value
PaO_2_ [kPa]	≥6.85	20.5%	1	/	Excluded for multicollinearity **	
	<6.85	52.4%	4.282 (2.770–6.619)	<0.001 *		
SaO_2_ [kPa]	≥87.5	18.9%	1	/	1	/
	<87.5	56.9%	5.670 (3.635–8.844)	<0.001 *	5.120 (2.758–9.505)	<0.001 *
Lymphocyte count [10^9^/L]	≥1.20	14.9%	1	/	1	/
	<1.20	39.1%	3.662 (1.807–7.422)	<0.001 *	2.880 (1.218–6.809)	0.016 *
LDH [U/L]	≤793.5	24.8%	1	/	1	/
	>793.5	49.1%	2.934 (1.898–4.538)	<0.001 *	1.078 (0.580–2.002)	0.812
CK [U/L]	≤171	29.8%	1	/	1	/
	>171	45.0%	1.927 (1.257–2.955)	0.003 *	1.911 (0.978–3.733)	0.058
CRP [mg/L]	≤108.5	30.6%	1	/	1	/
	>108.5	40.9%	1.572 (1.039–2.376)	0.032 *	1.096 (0.594–2.024)	0.768
IL-6 [pg/mL]	≤59.75	21.6%	1	/	1	/
	>59.75	49.7%	3.586 (2.327–5.525)	<0.001 *	4.089 (2.136–7.826)	<0.001 *
Age [years]	≥68.5	29.8%	1	/	1	/
	<68.5	39.7%	1.554 (1.027–2.349)	0.037 *	1.976 (1.038–3.762)	0.038 *
Sex	Male	28.9%	1	/	1	/
	Female	45.7%	2.070 (1.361–3.147)	0.001 *	2.290 (1.158–4.529)	0.017 *
Need for oxygen therapy upon admission	No	13.8%	1	/	Excluded for multicollinearity **	
	Yes	36.8%	3.635 (1.239–10.661)	0.019 *		
BMI	<30	22.1%	1	/	Excluded for multicollinearity **	
	≥30	56.5%	4.568 (2.955–7.060)	<0.001 *		
BF%	Normal/High	13.2%	1	/	1	/
	Very high	57.4%	8.892 (5.439–14.538)	<0.001 *	8.059 (3.990–16.276)	<0.001 *
VF	Normal/High	27.2%	1	/	1	/
	Very high	47.8%	2.448 (1.610–3.722)	<0.001 *	1.159 (0.566–2.372)	0.686
Time from disease onset [days]	/	/	0.923 (0.877–0.970)	0.002 *	0.941 (0.867–1.020)	0.140

Abbreviations: aOR—adjusted OR; ARDS—acute respiratory distress syndrome; BMI—body mass index; BF%—body fat percentage; CI—confidence interval; CK—creatine kinase; CRP—C-reactive protein; VF—visceral fat; IL-6—interleukin 6; LDH—lactate dehydrogenase; OR—odds ratio; PaO_2_—partial pressure of oxygen; SaO_2_—oxygen saturation of the blood. * Statistical significance level at <0.5. ** Variables excluded due to the multicollinearity principle of multiple logistic regression.

## Data Availability

The data presented in this study are available on request from the corresponding author.

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
