# Peer review of "Anthropometric Measurements and Admission Parameters as Predictors of Acute Respiratory Distress Syndrome in Hospitalized COVID-19 Patients"

_biomedicines, 2023, doi:10.3390/biomedicines11041199_

Round 1

Reviewer 1 Report

1) Abstract.  Conclusion: BIA measurements and laboratory markers upon admission can be helpful in determining the risk of ARDS in COVID-19 patients. Please ameliorate the conclusion and underline the novelty of the study.

2) 1. Introduction. L 52-59. Despite continuous research efforts, the exact mechanisms of clinical deterioration  are not fully elucidated. In a practical manner, it would be beneficial to single out condi-  tions and clinical characteristics associated with ARDS development in order to construct  prediction models and timely single out patients at risk of clinical deterioration. Accord-  ing to the available literature, many parameters are associated with COVID-19 ARDS, such as older age, male sex, certain comorbidities, smoking, impaired gas exchange, ele-  vated biomarkers of inflammation, impaired coagulation, and others [5-11]. However, the  exact selection of predictors and their impact on ARDS development in hospitalized  COVID-19 patients are not uniform across the literature, mostly due to the significant het-erogeneity of cohort characteristics and methodological approach. In order to discuss the previously described points, important references are needed to be added, such as:

a-Long-Term Impact of COVID-19: A Systematic Review of the Literature and Meta-Analysis. Biomedicines 20219, 900. https://doi.org/10.3390/biomedicines9080900

b-Different Methods to Improve the Monitoring of Noninvasive Respiratory Support of Patients with Severe Pneumonia/ARDS Due to COVID-19: An Update. J Clin Med. 2022 Mar 19;11(6):1704. doi: 10.3390/jcm11061704.

c- COVID-19 and Post-Acute COVID-19 Syndrome: From Pathophysiology to Novel Translational Applications. Biomedicines 202210, 47. https://doi.org/10.3390/biomedicines10010047

3) Introduction. L 68-72. For those reasons, we aimed to single out predictors of ARDS in hospitalized  COVID-19 patients, based on parameters available in the first 24 hours upon admission.  Furthermore, we aimed to examine the impact of obesity, accessed through the bioelectri-  cal impedance (BIA) method, on COVID-19 ARDS development. To the best of the au-  thors’ knowledge, this is the first registry regarding BIA measurements and COVID-19  outcomes. Improve the description of study aim.

4) 3. Results. Underline in the manuscript the most important statistically significant data to support the results.

5) Figure 1. Body composition categories for BMI (1a), %BF (1b), and VF (1c), in regard to ARDS development. Please improve the quality of this figure.

6) L 313-320. Our study had certain limitations. Firstly, COVID-19, analog to other infectious and  inflammatory conditions, impacts body composition. In order to minimize that effect,  we measured patients within the first 48 upon admission. In addition, although BIA  measurements provide satisfactory insight into total body fat and fat-free mass, this  method has difficulties in distinguishing visceral from abdominal fat, for which CT and  MRI remain the gold standard [34, 46]. However, the primary aim of the study was to  examine the possibility of BIA measurements upon admission, with all its limitations, to  predict ARDS development. Secondly, the number of patients included may be insuffi-cient for the generalization of the results. Improve the description of the limits of the study and the possible clinical implications. 

Author Response

Dear Reviewer, thank you for your constructive suggestions. I gave my best to answer all the comments and suggestions. Hope that we have met your standards and sufficiently improved the manuscript.

The corrections are indicated in the comments and highlighted (light gray) in the text.

  1. The "conclusion" section in the abstract is now corrected.
  2. The suggested references are added. (References 47-49)
  3. The study aims are rephrased. (Lines 69-75)
  4. Results - I am afraid I did not clearly understand the suggestion. All the statistical and numerical values are given in the tables, and they were not included in the text in order to reduce the text burden and increase comprehensibility. Instead, we decided to highlight and thoroughly comment on the results of the discussion. If, in your opinion, the results still need corrections in order the fulfill the publication's standards, I would have to politely ask you to help me with perhaps more detailed instructions on how to improve this section.
  5. The figure quality is improved (initial converting into 600 dpi actually negatively affected the pixels, and it is now improved with 300 dpi) (Line 199)
  6. The description of study limitations is hopefully improved, and the potential clinical implications are included. (Lines 324-335)

Reviewer 2 Report

Reviewer comments and suggestions

The authors in this study utilized the anthropometric measurements and admission parameters as predictors of acute respiratory distress syndrome (ARDS) in hospitalized COVID-19 patients. The study included a prospective cohort study was conducted on 407 consecutive COVID-19 patients hospitalized at the University Clinical Center Kragujevac between September 2021 and March 2022. Body composition was evaluated using the BMI, body fat percentage (BF%), and visceral fat (VF) via bioelectrical impedance analysis. The result of the study showed that patients with BMI above 30 kg/m2, very high BF%, and/or very high VF levels were at a significantly higher risk of developing ARDS compared to non-obese patients (OR 4.568, 8.892, and 2.448, respectively). The study concluded that BIA measurements and laboratory tests upon admission could be supportive in determining the risk of ARDS in COVID- 19 patients

Overall, the manuscript was well written. However, a few concerns/comments needed to be explained/modified. 

  1. Line 21 ARDS It should be in full form as first time used this term
  2. Line 34 Add (BIA)
  3. Line 51 Could you please describe the term “acute respiratory distress syndrome” its medical condition
  4. Line 100-104 Please provide the data in details also ethical approval number is also needed.
  5. Line 127 method of measuring VF and BF should also be mentioned
  6. Line 157 what does it mean, please explore.
  7. Comments for Figure 1 a Euhtrophic, can we use normal in place of this. I think it should be consistent.
  8. Comments for figure 2 ROC analysis should be discussed well in the result part.
  9. Line 243 The authors did not discuss any relationship BF and the above-discussed points
  10.  Line 254 What would be the possible reason for this?
  11.  Line 270 Please mention the significance of these tests for the common reader of your manuscript
  12.  Line 300-301 Please explain these results with the help of the ROC curve that the authors mentioned in the MS.

Author Response

Dear Reviewer, thank you for your constructive suggestions. I gave my best to answer all the comments and suggestions. Hope that we have met your standards and sufficiently improved the manuscript.

The corrections are indicated in the comments and highlighted (light gray) in the text.

  1. The ARDS is written in full. I apologize for such an oversight. (Line 28)
  2. The abbreviation is now used. (Line 29)
  3. The ARDS is now clinically described, before giving the Berlin definition criteria. (Lines 85-87)
  4. The supplementary table was created in order to improve the understanding of patient exclusion and the enrolment process. The Ethical committee approval is now also mentioned (details are provided in the: "Institutional Review Board Statement"). (Lines 81, 82, 94)
  5. BIA method of measuring BF% and VF is now included. (Lines 125, 126)
  6. The absence of comorbidities' impact on ARDS is now discussed. I hope it is satisfactory. (Lines 249-260)
  7. Euthropic is replaced with normal weight. (Line 199)
  8. The C-index, as well as the model's characteristics are now implemented in the discussion. (Lines 303-305)
  9. The relationship between body fat and inflammation is now implemented. (Lines 236-241)
  10. Corrected as part of comment #6. (Lines 249-260)
  11. The significance of SaO2 and PaO2 is now discussed. (Lines 278-280)
  12. The diversity of the model found in the literature is now discussed from the statistical and methodological point of view. (Lines 309-313)